# Alterations of the Endoplasmic Reticulum (ER) Calcium Signaling Molecular Components in Alzheimer’s Disease

**DOI:** 10.3390/cells9122577

**Published:** 2020-12-01

**Authors:** Mounia Chami, Frédéric Checler

**Affiliations:** Team Labelled “Laboratory of Excellence (LABEX) DistAlz”, INSERM, CNRS, IPMC, Université Côte d’Azur, 660 route des Lucioles, Sophia-Antipolis, 06560 Valbonne, France; checler@ipmc.cnrs.fr

**Keywords:** calcium, Alzheimer’s disease, endoplasmic reticulum, SERCA, IP_3_R, RyR, S1T, presenilin

## Abstract

Sustained imbalance in intracellular calcium (Ca^2+^) entry and clearance alters cellular integrity, ultimately leading to cellular homeostasis disequilibrium and cell death. Alzheimer’s disease (AD) is the most common cause of dementia. Beside the major pathological features associated with AD-linked toxic amyloid beta (Aβ) and hyperphosphorylated tau (p-tau), several studies suggested the contribution of altered Ca^2+^ handling in AD development. These studies documented physical or functional interactions of Aβ with several Ca^2+^ handling proteins located either at the plasma membrane or in intracellular organelles including the endoplasmic reticulum (ER), considered the major intracellular Ca^2+^ pool. In this review, we describe the cellular components of ER Ca^2+^ dysregulations likely responsible for AD. These include alterations of the inositol 1,4,5-trisphosphate receptors’ (IP_3_Rs) and ryanodine receptors’ (RyRs) expression and function, dysfunction of the sarco-endoplasmic reticulum Ca^2+^ ATPase (SERCA) activity and upregulation of its truncated isoform (S1T), as well as presenilin (PS1, PS2)-mediated ER Ca^2+^ leak/ER Ca^2+^ release potentiation. Finally, we highlight the functional consequences of alterations of these ER Ca^2+^ components in AD pathology and unravel the potential benefit of targeting ER Ca^2+^ homeostasis as a tool to alleviate AD pathogenesis.

## 1. Introduction

### 1.1. Ca^2+^ Signaling

As a signal transduction molecule, calcium (Ca^2+^) regulates a large number of neuronal processes including growth and differentiation, neurotransmitter release and synaptic function, activity-dependent changes in gene expression and apoptosis [1]. Cytosolic Ca^2+^ ([Ca^2+^]cyt) signals are regulated in a spatiotemporal-dependent manner underlined by an intricate interplay between Ca^2+^ entry through the plasma membrane, storage in the internal stores (i.e., the endoplasmic reticulum (ER), considered the major dynamic Ca^2+^ intracellular pool), Ca^2+^ mobilization from the ER and its buffering by Ca^2+^-binding proteins (CaBP) (Figure 1). Ca^2+^ entry through the plasma membrane occurs through ligand-dependent Ca^2+^ receptors (i.e., *N*-methyl-d-aspartate receptor (NMDA) and Alpha7 nicotinic acetylcholine receptors (nAChRs)) and through voltage-gated Ca^2+^ channels (VGCC) (Figure 1). Ca^2+^ mobilization from the ER occurs through activation of the inositol 1,4,5-trisphosphate receptors (IP_3_R) downstream of metabotropic receptors (Figure 1), or through the activation of ryanodine receptors (RyRs) that are activated by a slight increase in [Ca^2+^]cyt, a mechanism known as Ca^2+^-induced Ca^2+^ release (CICR) (Figure 1). Elevations of cytosolic Ca^2+^ signals are “shut down” through the plasma membrane Na^+^/Ca^2+^ exchanger (NCX) and two Ca^2+^ ATPases which consume ATP to actively extrude Ca^2+^ out of the cells (i.e., the plasma membrane Ca^2+^ ATPase (PMCA)) or to actively sequester Ca^2+^ into the ER lumen (i.e., the sarco-endoplasmic reticulum Ca^2+^ ATPase (SERCA)) (Figure 1). Intriguingly, coupling between ER Ca^2+^ depletion and Ca^2+^ influx through the plasma membrane occurs through a canonical store-operated Ca^2+^ entry (SOCE) pathway [2] mainly consisting of a direct physical interaction between the Ca^2+^-sensing stromal interacting molecules (STIM1/2) oligomers within the ER membrane and the pore-forming ORAI proteins in the plasma membrane [3,4,5] (Figure 1). Several lines of evidence indicate that Ca^2+^ homeostasis could be disrupted upon cellular challenges as well as in neurodegenerative conditions.

### 1.2. Alzheimer’s Disease

Alzheimer’s disease (AD) is an age-associated dementia disorder characterized by the accumulation of extracellular amyloid-beta (Aβ) peptides in the senile plaques and by the hyperphosphorylation of tau (pTau) protein, leading to intracellular protein aggregation into bundles or filaments that are deposited as neurofibrillary tangles [6,7,8]. Notably, Aβ peptide derives from the sequential processing of the β-amyloid precursor protein (βAPP referred to as APP hereafter) [9,10] by the β-seceretase (BACE1) and the γ–secretase complex (composed of presenilins (PSs: PS1 or PS2, the catalytic subunits of the enzyme), Nicastrin, anterior pharynx-defective-1 (APH-1) and presenilin enhancer-2 (PEN-2) [11,12])) (Figure 2). Importantly, a significant number of aggressive AD cases generally characterized by early onset are inherited in an autosomal-dominant manner (FAD: familial AD) and are caused by mutations on APP and on PS1 and PS2 [13,14] (Figure 2). These mutations either modify the nature of Aβ peptides and/or affect the levels of their production [15,16]. Besides the canonical disease-associated intracellular pTau and extracellular Aβ accumulations, recent studies unraveled additional processes that could contribute to AD progression, including: (i) the intracellular accumulation of Aβ [17,18] and other APP-derived fragments [18,19,20,21,22,23,24], and (ii) the spreading of both extracellular Tau and Aβ between neurons and between neurons and glial cells [25,26].

### 1.3. Physiology of ER Calcium Handling in Neurons

The ER forms a continuous and highly motile network distributed throughout the neuron. Within dendrites and dendritic spines, ER Ca^2+^ release is involved in modulating postsynaptic responses and synaptic plasticity [27]. In presynaptic nerve terminals, as well as in growth cones, ER is involved in vesicle fusion and neurotransmitter release [28,29]. In the soma, ER Ca^2+^ handling is coupled to the activation of Ca^2+^-sensitive kinases and phosphatases [30]. In the perinuclear space, ER Ca^2+^ handling triggers gene transcription [31]. Ca^2+^ mobilization from the ER has been shown to be involved in growth cone activity and in the formation of new connections and/or the strengthening of preexisting connections that occur during learning and memory in the adult brain [32].

### 1.4. Calcium Deregulation in AD

As stated above, the tight but subtle control of intracellular Ca^2+^ homeostasis is required for neuronal health, development and function [29,30,33,34]. Therefore, persistent imbalance in Ca^2+^ entry and clearance alters cellular integrity, leading to cellular homeostasis disequilibrium. These Ca^2+^ deregulations ultimately trigger excessive proliferation or cell death depending on the strength and the duration of the insult and in a cell-type-specific manner. Ca^2+^ signaling deregulation has a central role in AD pathophysiology [35]. The relevance of Ca^2+^ signaling in AD is supported by the fact that Ca^2+^ alterations were reported in both sporadic (SAD) and familial (FAD) forms of AD and that this can exacerbate Aβ formation and promote tau hyperphosphorylation [35,36,37]. As first evidence, in vitro studies have shown that Ca^2+^ may directly interact and enhance the proteolytic activity of BACE1 [38] and to stabilize γ–secretase and enhance its activity in reconstituted in vitro assay [39]. Moreover, tau hyperphosphorylation at disease-specific sites has been associated with abnormal intracellular Ca^2+^ signaling occurring upstream of Ca^2+^/calmodulin (CaM)-dependent protein kinase II (CaMKII) and CDK5 activation [37,40,41,42]. The bulk of data gathered these last 30 years allows us to draw up a scenario where Ca^2+^ deregulation is not only a consequence of the disease but also participates in a feedback loop to disease progression and amplification [35,36,43,44,45,46]. These studies reported a Ca^2+^-dependent enhancement of APP processing and the production of toxic APP-derived fragments, activation of signaling cascades through the modulation of kinases and phosphatases activities, thus affecting synaptic plasticity and cognitive function [34,35,36,47,48,49].

Several studies demonstrated a tight relationship between altered Ca^2+^ handling and the amyloidogenic cascade. These studies lead to identifying the physical or functional interaction of Aβ with several Ca^2+^ handling proteins in various AD models. At least four lines of evidence have emerged: (i) at the plasma membrane, Aβ has been shown to form a cation channel [50], or to act as a channel-modulator for the VGCCs, the nAChRs, the ionotropic glutamatereceptors NMDARs and AMPARs (α-amino-3-hydroxy-5-methyl-4-isoxazolepropionic acid receptors), the Ca^2+^ homeostasis modulator 1 (CALHM1), and more recently the store-operated Ca^2+^channels (SOCE) (Figure 2) [51,52,53,54,55,56,57,58,59,60]; (ii) dysfunctional mitochondria were associated with Aβ-mediated Ca^2+^ toxicity [61,62] (discussed in this Special Issue [63]). Importantly, mitochondrial permeability transition pore, mitochondrial Ca^2+^ uniporter (MCU) dysfunctions and impaired mitochondrial Ca^2+^ efflux contribute to mitochondrial alteration in AD [63,64,65]; (iii) the autophagic failure in AD has been linked to lysosomal degradation defects [24,66] likely occurring upon lysosomal Ca^2+^ depletion [67,68]; (iv) a complex scenario of AD-associated ER Ca^2+^ dysregulation also emerged, where disturbances were linked to presenilin (PS1 and PS2)-associated ER Ca^2+^ leak and/or ER Ca^2+^ release potentiation functions [69,70,71,72,73], dysfunctional SERCA activity [74] and the upregulation of the recently described SERCA1 truncated isoform (S1T) [75], alterations of IP_3_Rs function [56,69,70,72,76,77,78,79,80,81] and dysfunctional RyRs [44,80,82,83,84,85,86,87,88,89,90,91,92,93,94] (Figure 2).

Besides APP-derived amyloidogenic products, previous studies described a physiological role of APP in regulating Ca^2+^ signaling. Knockdown of endogenous APP increases the frequency and reduces the amplitude of neuronal Ca^2+^ oscillations [95]. In addition, a recent study specifically reported that APP-deficient cells exhibited elevated resting Ca^2+^ levels in the ER and reduced ER Ca^2+^ leakage rates [96]. Pathogenic tau has also been associated with nuclear Ca^2+^ deregulation [97], with increasing the ionic current of artificial membranes [98], with inducing spontaneous Ca^2+^ oscillations in the neurons [99] and with the inhibition of mitochondrial Ca^2+^ efflux via the mitochondrial Na^+^/Ca^2+^ exchanger [99] (also discussed in this Special Issue [63]).

In this review, we will specifically present an update of the alterations of the molecular components controlling ER Ca^2+^ signaling in AD and discuss the potential benefit of targeting ER Ca^2+^ homeostasis as a tool to alleviate AD pathogenesis.

## 2. The Ryanodine Receptors: RyRs

RyRs are a family of three mammalian isoforms, RyR1, RyR2 and RyR3, mainly expressed in the skeletal muscle, heart and brain. All RyRs isoforms are expressed in the brain, with an abundance range of order as follows, RyR2 > RyR1 >> RyR3 [100,101]. RyRs activity is influenced on the one hand by Ca^2+^, Mg^2+^ and ATP [102,103,104,105] and, on the other hand, by the integrated effects of co-proteins forming RyR1 and RyR2 homotetramer macromolecular complexes [106,107,108]. These include calmodulin (CaM) [109,110], FKBP12 (12.0 kDa) and FKBP12.6 (12.6 kDa), known as Calstabin1 (Cal1) and Calstabin2 (Cal2), respectively [111]; PKA anchored to RyR1 and RyR2 via a kinase anchoring protein (mAKAP) [112,113], and Ca^2+^/calmodulin-dependent protein kinase II (CaMKII) [114]. Other regulatory proteins were also described to interact with RYR1, thus controlling the channel gating activity [109]. RyR1/2 macromolecular complexes contain also the requisite molecular machinery allowing channel dephosphorylation (i.e., PP1 and PP2A) [113,115,116].

Enhanced RyR-mediated Ca^2+^ release was reported in primary cultured neurons derived from 3xTg-AD mice (knock in (KI) for the mutated PS1M146V and overexpressing mutated APP and microtubule-associated tau protein (PS1M146V/APPswe/tauP301L)) [85,87]. This was further confirmed in cellular models expressing wild-type or mutated APP, PS1 or PS2 [44,80,82,83,84,86,87,88,89,90,91,92,93,94,117]. Exacerbated IP_3_R-evoked Ca^2+^ signals in AD mice (PS1KI and 3xTg-AD)-derived neurons were shown to be linked to RYR-associated CICR [85]. These findings were further supported by using the RyR blocker dantrolene (Dant), shown to reduce enhanced [Ca^2+^]cyt level [92,93,118]. While some studies reported that RyR dysfunction in AD-related study models occurs independently of PS mutation or overexpression, namely in models expressing APP and overproducing Aβ [86,92,119,120,121,122], in many cases, PS mutation-mediated Ca^2+^ deregulation was associated with the alteration of the activity of RyRs (discussed beyond in PSs chapter). In addition, it was also reported that exogenous Aβ oligomers may directly stimulate RyR-mediated Ca^2+^ release [123] and that the application of soluble Aβ caused a marked increase in channel open probability [124].

RyR isoform expression is modified throughout AD progression and between different brain regions [125]. Exogenous application of Aβ peptide was also shown to specifically increase RyR3 isoform expression [86,123]. RyRs mRNAs increase throughout the lifetime of PS1-M146V transgenic mice and 3xTg-AD mice [84,85,87] as well as in cellular and mice AD models overexpressing wild-type or mutated APP (bearing the Swedish mutation APPswe) [92]. Conversely, neuronal conditional PS1/2 knockout (KO) (PScDKO) is associated with a downregulation of RyR2 expression, demonstrating that PS may regulate Ca^2+^ homeostasis and synaptic function via RyRs [126]. It has been proposed that the modulation of RyR expression may act as a disease promoter or a compensatory beneficial mechanism. In fact, while on the one hand, enhanced [Ca^2+^]cyt response is associated with the increased expression of RyRs [127], the activation of the ER stress response factor X-box binding protein 1 spliced isoform (XBP1s) may occur upon Aβ oligomer treatment [128,129], triggering a reduction of [Ca^2+^]cyt linked to the down-expression of the RyR3 isoform [130]. Accordingly, a dual role for endogenous RyR3 has been suggested in an AD mouse model. Thus, the deletion of RyR3 in young (≤ 3 mo) APPPS1 mice increased hippocampal neuronal network excitability and accelerated AD pathology, leading to mushroom spine loss and increased Aβ accumulation. Meanwhile, deletion of RyR3 in older APPPS1 mice (≥6 mo) rescued network excitability and mushroom spine loss, reduced Aβ load and reduced spontaneous seizure occurrence [131] (Figure 2).

RyRs mutations are liked to various pathologies affecting muscle and heart [132,133]. The development of transgenic mouse models (i.e., KO of RyR1, RyR2 or RyR3, or expressing RyR harboring disease mutations, or lacking exon sequence) [134] strengthens the fact that RyRs play a key role in physiology and pathophysiology. The viability of RyR3 KO mouse, in contrast to the RyR1 and RyR2 KO mice [135,136], led to the demonstration that RyR3-deficient mice exhibit decreased social behavior [137], greater locomotor activity [136,138], altered memory [138,139] associated with impaired maintenance of long-term potentiation (LTP) [140]. To date, no mutations have been reported in RYRs linked to brain disorders. Nevertheless, the role of leaky RyR2 in the pathogenesis of epilepsy has been described in the RyR2-R2474S mice model [101]. Interestingly, three single nuclear polymorphisms were significantly associated with risk for hypertension, diabetes and AD [141]. A meta-analysis based on four genome-wide association study (GWAS) also identified *RYR3* association with AD risk [142]. Another study observed a significant interaction between *RYR3* and *CACNA1C* (gene encoding for the Ca^2+^ voltage-gated channel subunit Alpha1 C) in three independent datasets of AD Neuroimaging Initiative cohorts [143].

RyRs post-translational modifications (PTMs) shift the channel from a finely regulated state to a non-regulated Ca^2+^ leak channel. RyR PTMs were associated with different pathologies affecting skeletal muscle, heart and, recently, brain [133] [113,144,145,146,147,148,149,150]. Experimental transgenic mice expressing RyR harboring PKA-non-phosphorylated sites or phosphomimetic RyR mutants demonstrated the role of the PKA phosphorylation site in RyR macromolecular complex remodeling, Calstabin dissociation and ER Ca^2+^ leak [133]. In addition to phosphorylation sites, RyRs also contain a large number of amino acid residues that are potential targets for reactive oxygen species (ROS) and for reactive nitrogen species (RNS) [108,151,152]. Recently, we described a new molecular mechanism and signaling cascade underlying altered RyR-mediated intracellular Ca^2+^ release in AD [116,150,153]. We reported that the RyR2 channel undergoes PKA phosphorylation, oxidation/nitrosylation and depletion of the channel stabilizing subunit Calstabin2 in SH-SY5Y neuroblastoma cells expressing APP harboring the familial Swedish mutations (APPswe), in APP/PS1 (APPswe, PS1-M146V), as well as in 3xTg-AD, transgenic mice models and, most importantly, in human SAD brains [150,153]. We further reported that RyR2 macromolecular complex remodeling occurs through synergistic mitochondrial reactive oxygen species (ROS) production and β-adrenergic stimulation [150,153]. Notably, oxidative stress is considered a major contributor to AD pathogenesis [154,155], and β2-adrenergic receptors (β2-ARs) have also been implicated in the development of AD [51,156,157,158,159,160,161]. However, targeting β-adrenergic signaling is questionable, since both beneficial versus defective effects were described in AD mice [162,163,164]. In our study, we specifically targeted the downstream PKA-mediated RyR2 phosphorylation and macromolecular complex destabilization (Figure 2). We showed that pharmacological stabilization of calstabin2 on the RyR2 macromolecular complex by S107 (a benzothiazepine derivative molecule [101]) reduces elevated Ca^2+^ signals in AD cells [153], prevents ER Ca^2+^ leakage and reduces single channel open probabilities in AD mice brains [150]. Most importantly, S107 treatment reduces APP processing and Aβ production both in vitro and in vivo [150,153]. S107 administration also inhibited calpain activity and AMPK-dependent tau phosphorylation in an APP/PS1 mouse model [150]. These data agree well with previously reported studies demonstrating the beneficial effects of the pharmacological targeting of RyR with dantrolene [88,92,93,165]. In support of these findings, RyR macromolecular complex stabilization improved the hippocampal synaptic plasticity (LTP and LTD) and cognitive function of APP/PS1 and 3xTg-AD mice [150]. Importantly, we further showed that crossing APP/PS1 mice with RyR2-S2808A KI mice, harboring RyR2 channels that cannot be PKA-phosphorylated, resulted in improved cognitive function and decreased neuropathology. In contrast, phosphormimetic RyR2-S2808D KI mice exhibit early altered hippocampal synaptic plasticity (LTP and LTD) and cognitive dysfunction [150]. Overall, these results emphasize the broad implication of RyRs in ER Ca^2+^ signaling deregulation in AD occurring through the regulation of RYRs expression, CICR-dependent activity, macromolecular complex stability-linked to β2-AR signaling cascade, Aβ- and PS-mediated RyRs channel opening and likely RYR3 gene polymorphism.

## 3. The Inositol 1,4,5-Trisphosphate Receptors: IP_3_Rs

Among the three IP_3_Rs isoforms, the predominant one in neurons is IP_3_R1 [166,167,168,169]. In addition to Ca^2+^ and IP_3_, there are other allosteric IP_3_R modulators, including ATP [170]. The activity of IP_3_R can also be regulated by its phosphorylation by different kinases [166,170]. Among them are PKA, protein kinase C (PKC), cGMP-dependent protein kinase (PKG), CaMKII and different protein tyrosine kinases. Moreover, similarly to RyRs, the IP_3_Rs can also be regulated by the redox status and by several interacting proteins (i.e., CaM-related Ca^2+^-binding proteins (CaBPs), Bcl2 family members, proteases (Caspase-3 and calpain) and ER lumen-specific protein (ERp44) [170]).

IP_3_Rs activity controls spine morphology, synaptic plasticity and memory consolidation [171,172,173]. Notably, alterations of IP_3_Rs expression and function were reported to be implicated in Ca^2+^ signaling deregulation in several AD models [174]. Ca^2+^ imaging experiments demonstrated that orthologous expression of FAD PS1 mutants potentiates IP_3_-mediated Ca^2+^ release [175]. These data were confirmed in cortical neurons isolated from PS1-M146V KI mice [79] and in cells expressing FAD PS1-DeltaE9 mutant [176]. PS1-DeltaE9 mutant cells harbored enhanced basal phosphoinositide hydrolysis and cyt[Ca^2+^], which were both reversed by the PLC inhibitor neomycin. PS1-DeltaE9 mutant cells also showed high basal [Ca^2+^] and agonist-evoked Ca^2+^ signals that were reversed by xestospongin C (XeC, a reversible IP_3_R antagonist) [176]. The molecular mechanisms underlying enhanced IP_3_R-mediated Ca^2+^ release have been described to be PS-dependent and/or PS-independent [73,177] (also discussed in PSs chapter below). The computational modeling of single IP_3_R activity was used to analyze and quantify the pathological enhancement of IP_3_R function by FAD-causing mutant PS [178]. This study revealed that the gain-of-function enhancement of IP_3_R was sensitive to both IP_3_ and Ca^2+^, thus triggering a higher frequency of local Ca^2+^ signals, while enhancing the activity of the channel at extremely low ligand concentrations will lead to spontaneous Ca^2+^ signals in cells expressing FAD-causing mutant PS [178]. It has been consequently observed that the gain-of-function enhancement of IP_3_R channels in cells expressing PS1-M146L leads to the opening of mitochondrial permeability transition pore (PTP) in high-conductance state, triggering a reduction in the inner mitochondrial membrane potential and in NADH and ATP levels [179]. Conversely, genetic reduction of IP_3_R1 normalizes disturbed Ca^2+^ signaling in PS1-M146V KI mice and most importantly alleviates AD pathogenesis (i.e., rescues aberrant hippocampal long-term potentiation (LTP), attenuates Aβ accumulation and tau hyperphosphorylation and memory deficits) in both PS1-M146V KI and 3xTg-AD mice [72]. Accordingly, in vitro experiments showed that XeC effectively ameliorated Aβ42-induced apoptosis and intracellular Ca^2+^ overload in the primary hippocampal neurons [180]. Notably, intracerebroventricular injection of XeC reduced the number of Aβ plaques, alleviated ER stress response and significantly improved the cognitive behavior of APP/PS1 mice [180]. Exacerbated IP_3_R-mediated Ca^2+^ release is also linked to Aβ, independently of PS overexpression/mutation. Jensen L.E., et al. reported that Aβ42 induced elevation of cytosolic Ca^2+^ in an IP_3_R-dependent and -independent manner [91]. In addition, it was also shown that the treatment with Aβ42 significantly increased mRNA levels of IP_3_R1/2 and mGluR5 [181]. Enhanced IP3R1 expression and ER Ca^2+^ release were also reported in astrocytes derived from the entorhinal cortex and from the hippocampus from WT mice and mice treated with Aβ42 oligomers [182]. Finally, as stated above, IP_3_R function is regulated by several binding proteins. Thus, it is also conceivable that any alteration of the expression, localization, activity and binding affinity of these proteins may affect IP_3_R structural/functional state, thus impacting AD development.

## 4. Presenilins 1 and 2: PS1/2

PS1 and PS2 are multispanning transmembrane (TM) proteins located in intracellular membranous organelles such as the ER, nuclear envelope and Golgi apparatus but also in multiple secretory and endocytic organelles as well as the plasma membrane [183]. PS1 was first cloned as a causative gene of FAD [184]. Its homologue PS2 gene was then identified, sharing an approximately 60% sequence homology as a whole and approximately 90% within the TM domains [185,186]. Accordingly, PS1 and PS2 were also shown to share similar predicted topology [187,188,189].

Both PS1 and PS2 are expressed in neurons [190] and are essential for embryonic development since PS1 KO mice die at birth [191] and PS1/PS2 double KO mice (PSDKO) mice die before embryonic day 9.5 [192]. Importantly, PSs were also shown to play key roles in neuronal function and survival. Therefore, conditional PSDKO mice show impaired spatial and associative memory, deficits in short- and long-term plasticity [193,194] and develop synaptic, dendritic and neuronal degeneration in an age-dependent manner [193]. Importantly, being the catalytic component of γ–secretase complex cleaving the APP [195], most of the PS mutations associated with early-onset FAD affect APP processing and, more particularly, the ratio Aβ40/42 by increasing the aggregation-prone Aβ42 species [196,197,198]. Several studies proposed the contribution of PSs to ER Ca^2+^ signaling deregulation in AD. It has been proposed that PSs act as ER Ca^2+^ leak channels, and FAD mutations in PSs disrupt this function, leading to ER Ca^2+^ overload [69,70,76]. Tu et al. also proposed that the full-length PSs function as ER Ca^2+^ leak channels independently of other γ–secretase components [69]. Cysteine point mutants combined with NMR studies revealed that TM7 and TM9, but not TM6, could play an important role in forming the conductance pore of mouse PS1 [77]. A recent study investigated the interaction of Ca^2+^ with both PS1 and PS2 using all-atom molecular dynamics (MD) simulations in realistic membrane models [199]. Although the Ca^2+^ leak event linked to PS1 or PS2 has been challenged in this study, the obtained data demonstrated the presence of four Ca^2+^ sites in membrane-bound PS1 and PS2 [199]. The authors speculated that Ca^2+^ may prevent PS maturation (i.e., “presenilinase” endoproteolysis generating PS N-and PS C-terminal derivatives [200]) by triggering conformational changes, thus preserving the immature Ca^2+^ regulation function. Meanwhile, conversely, PS maturation yielding a biologically active PS would abolish this Ca^2+^-regulatory function [199]. Nevertheless, the PS-associated Ca^2+^ leak function was discussed in other studies proposing that FAD PSs directly potentiate the gating of IP_3_R [81]. Exaggerated IP_3_R-mediated Ca^2+^ responses were also reported in cells and neurons derived from transgenic mice expressing FAD-linked mutant PS1 or PS2 [84]. These findings agree well with data obtained in PS1-M146V KI mice neurons using whole-cell patch-clamp recording, flash photolysis and two-photon imaging [79]. Accordingly, genetic reduction of IP_3_R 1 normalizes disturbed Ca^2+^ signaling in FAD PS1 mice and alleviates AD pathogenesis in PS1-M146V KI mice [72]. Other studies point to PS-linked disruptions in RyR signaling as an important ER molecular component associated with enhanced ER Ca^2+^ signals in both 3xTg-AD and PS1-M146V (KI) neurons [80]. PS1/PS2 were also shown to harbor a physical interaction with RyRs in the ER [83,117,201]. Specifically, PS2 interacts with RyR and with sorcin (a RyR regulator) in a Ca^2+^-dependent manner in both cellular models and in the brain [201,202], thus increasing both mean currents and open probability of single brain RyR channels [203,204]. Discrepancies regarding the role of PSs in ER Ca^2+^ handling alterations were further highlighted in a recent study [205] showing that FAD PS2 mutants, but not FAD PS1, are able to partially block SERCA activity, thereby reducing ER Ca^2+^ content in either SH-SY5Y cells or FAD patient-derived fibroblasts [205]. Despite this incongruity concerning the exact molecular mechanism underlying PS-mediated ER Ca^2+^ deregulation, FAD PSs undoubtedly directly or indirectly contribute to the Ca^2+^ hypothesis in AD. However, whether PS-mediated ER Ca^2+^ deregulation is dependent on or independent of its endoproteolysis generating PS N-and PS C-terminal derivatives still remains an open question.

## 5. The Sarco-Endoplasmic Reticulum (SR/ER) Ca^2+^-ATPase and Its Truncated Isoform: SERCA and S1T

SERCAs are integral ER proteins preserving low [Ca^2+^]cyt by pumping free Ca^2+^ ions into the ER lumen, utilizing ATP hydrolysis. The SERCA pumps are encoded by three distinct genes (SERCA1-3), resulting in 12 known protein isoforms, with tissue-specific expression patterns. SERCA2b is the most expressed isoform in neurons [206]. Despite the well-established structure and function of the SERCA pumps, their role in the central nervous system and whether it could be affected in brain diseases remain to be definitely established. Interestingly, SERCA-mediated Ca^2+^ dyshomeostasis has been associated with neuropathological conditions, such as bipolar disorder, schizophrenia, Parkinson’s disease but also AD [207]. An initial study showed that SERCA activity is reduced in fibroblasts isolated from PSDKO. Immunoprecipitation analyses suggested a physical interaction between SERCA and PS1 and PS2 [74] and that modulation of SERCA expression regulates Aβ levels [74]. The interaction of PS1 holoprotein was further demonstrated in cells overexpressing PS1 and subjected to tunicamycin treatment [208]. It has also been shown that overexpressed wild-type or mutated PS2 triggered ER-passive leakage through IP_3_R and RyR but also potently reduced ER Ca^2+^ uptake, an effect that has been counteracted by the overexpression of SERCA2b [71]. A recent study reported that the pharmacological SERCA activation by a quinoline derivative (CD1163), discovered via high-throughput screening of small molecules library, provides some beneficial effects in APP/PS1 mice [209] (Figure 2). Overall, these studies pinpointed the potential contribution of SERCA to ER Ca^2+^ dyshomeostsis in AD cellular study models. However, dedicated studies in mice AD models and in human-derived samples are still needed to further support the beneficial versus pathogenic role of the modulation of SERCA expression and activity in AD development.

Accumulation of unfolded proteins into the ER as well as alteration of ER Ca^2+^ homeostasis induce ER stress, eliciting an unfolded protein response (UPR) [210,211]. Several studies have reported that UPR occurs in human AD brains [212,213] and in several AD study systems [214,215,216]. We previously demonstrated that the human SERCA1 truncated isoform (S1T) [217] is induced under pharmacological and physiopathological ER stress through the activation of the PERK-eIF2α-ATF4-CHOP pathway [218]. In turn, S1T expression induction triggered an amplification of ER stress and mitochondrial apoptosis [218]. UPR activation has been proposed to be linked to intracellular Aβ accumulation [219]. In a recent study, we revealed that S1T is upregulated in SH-SY5Y cells expressing APPswe [75]. Importantly, biochemical data indicate that enhanced human S1T expression correlates with Aβ load in human AD-affected brains and that S1T high neuronal immunostaining is selectively observed in human AD cases harboring focal Aβ. We further demonstrated that S1T expression is induced by exogenous application of Aβ oligomers in cells [75]. Interestingly, S1T overexpression in return enhances APP processing and the production of APP-derived toxic fragments (APP C-terminal fragments and Aβ) in cells and in 3xTgAD mice. Mechanistically, we find that S1T-mediated elevation of APP proteolysis occurs through the upregulation of BACE1 expression and enhanced activity [75]. In agreement with these findings, several lines of evidence indicated that enhanced phosphorylation of PERK and eIF2α in the AD brain is associated with increased amyloidogenic APP processing [214,215,216] through increased BACE1 expression [220,221]. We have also to consider that BACE1 upregulation occurring downstream of [Ca^2+^]cyt elevation acts in a positive feedback loop with AD progression [222]. In addition, the induction of ER stress and the activation of UPR trigger neuroinflammation [223]. Accordingly, we demonstrated that S1T overexpression, as well as tunicamycin treatment, induce the expression of proinflammatory cytokines and increase the proliferation of active microglia [75]. Altogether, our data strengthen the molecular link between ER Ca^2+^ leak, ER stress and APP processing contributing to AD setting and/or progression.

## 6. The Molecular Bridge between ER Ca^2+^ Depletion and Plasma Membrane Ca^2+^ Entry: STIM/ORAI

The store-operated Ca^2+^ entry (SOCE) is an essential route for Ca^2+^ uptake to replenish intracellular Ca^2+^ stores [224]. The stromal interaction molecules STIM1 and STIM2 have been identified as essential components of SOCE and major sensors of the Ca^2+^ concentration located in the ER membrane (reviewed in [225]). Both STIM homologues are ubiquitously expressed in different cell types, with a higher STIM1 level in most tissues and a predominant expression of STIM2 in the brain [226]. A decrease in ER luminal Ca^2+^ concentration results in dissociation of Ca^2+^ from the STIM EF-hand domain, which, in turn, triggers oligomerization and activation of STIM1, a process that is reversed when luminal [Ca^2+^] returns to resting level [3,4]. Active STIM oligomers translocate to ER plasma membrane junctions and recruit and interact with ORAI channels located on the plasma membrane [227]. There are three ORAI isoforms displaying tissue-specific expression and activation patterns [225]. In addition, transient receptor potential channels (TRPC) can also be recruited by the ORAI/STIM complex, constituting an additional route for Ca^2+^ entry through the plasma membrane upon ER Ca^2+^ depletion [228] (Figure 1). Several TRPC isoforms were also identified and were described to harbor tissue/cell-specific expression and activation patterns [229].

Several studies pinpointed a role for STIM/ORAI in neuronal Ca^2+^ signaling-associated synaptic function [230,231]. Thus, the maturation of dendritic spines and the formation of functional synapses in immature hippocampal neurons is facilitated by the influx of Ca^2+^ through ORAI1 [232]. STIM also interact with/and or control the activity of several Ca^2+^ channels on the plasma membrane (L-type Ca^2+^ channels (Ca_V_1.2), L-type VGCCs and mGluR [232]). Several studies suggested that the disruption of neuronal SOCE underlies AD pathogenesis. A direct connection between Aβ-induced synaptic mushroom spine loss and the neuronal SOCE pathway was reported in two studies. Popugaeva et al. reported that the application of exogenous Aβ42 oligomers to hippocampal cultures or injection of Aβ42 oligomers directly into the hippocampal region resulted in the reduction of mushroom spines and activity of synaptic CaMKII, which were rescued by STIM2 overexpression [224]. Accordingly, similar findings were reported in APPKI hippocampal neurons accumulating extracellular Aβ42. Thus, it was shown that Aβ triggers mGluR5 receptor overactivation, leading to elevated ER Ca^2+^ levels, compensatory downregulation of STIM2 expression, impairment of synaptic SOCE and reduced CaMKII activity [58]. Inversely, overexpression of the constitutively active STIM1-D76A mutant and ORAI1 significantly reduced Aβ secretion [55]. A link between STIM1/2 and PS1 was also reported. The STIM2–SOCE–CaMKII pathway was downregulated in a PS1-M146V KI mouse model of AD, associated with loss of hippocampal mushroom spines [233], and conversely, STIM2 overexpression rescued synaptic SOCE and mushroom spine deficit in hippocampal neurons from PS1-M146V KI mice [233]. Intriguingly, even if STIM1 expression is not altered in the AD models cited above, it has been identified as a target of PS1-containing γ–secretase activity. In particular, FAD-linked PS1 mutations enhanced γ–secretase cleavage of STIM1, reducing the activation of ORAI1 and attenuating SOCE [234]. As a consequence, the inhibition of SOCE in hippocampal neurons triggered an alteration of the dendritic spine architecture [234]. A recent study showed that the hyperactivation of SOCE channels in neurons expressing PS1-DeltaE9 mutant is mediated by the STIM1 sensor and can be attenuated by pharmacological inhibition and genetic KO STIM1 [235]. Interestingly, SOCE in PS1-DeltaE9 mutant-expressing cells is not contributed by STIM2 but involves TRPC and ORAI subunits. Importantly, transgenic Drosophila flies expressing PS1-DeltaE9 in the cholinergic neuron system showed short-term memory loss, which was reversed upon pharmacological inhibition of STIM1 [235]. Accordingly, a recent study further supports the link between FAD PSs and altered SOCE, through the demonstration of reduced STIM1 expression in SH-SY5Y cells and in patient-derived fibroblasts expressing different FAD-PS mutations [205].

TRPC may also play a role in SOCE deregulation in AD. TRPC expression is not altered in mice and human AD brains [233]. However, reduced TRPC1 expression was observed in astrocytes derived from APP KO mice [181,236]. In addition, TRPC6 was shown to specifically interact with APP, thereby blocking its cleavage by γ–secretase and reducing Aβ production independently from its ion channel activity [237]. Conversely, PS2 mutations abolish agonist-induced TRPC6 activation [238]. Importantly, activation of TRPC6 stimulates the activity of the neuronal SOCE pathway in the spines and rescues mushroom spine loss and long-term potentiation impairment in APP KI mice [58]. A review by Prikhodko, V. et al. in this Special Issue addresses the potential use of TRPC modulators as drugs to treat AD [239].

## 7. Conclusions

Studies demonstrating the implication of ER Ca^2+^ deregulation in AD highlight a complex picture integrating several molecular ER Ca^2+^ components. This includes enhanced ER Ca^2+^ release through IP_3_R and RyR, dysfunctional ER Ca^2+^ uptake by SERCA and upregulation of the S1T truncated isoform, gain- or loss-of-function of PS components of the γ–secretase complex. Disease-associated remodeling of this Ca^2+^ machinery toolkit is also coupled to specific cellular signaling cascades modulating the activity (i.e., post-translational modifications, interactions with regulatory proteins) and/or the expression of these Ca^2+^ channels and pump (i.e., linked to ER stress). Several studies also pinpointed the direct interaction of Aβ peptide with several members of the ER Ca^2+^ machinery, thus contributing to ER Ca^2+^ dyshomeostasis. In addition, recent studies demonstrated that the failure of the SOCE molecular bridge between the ER and the plasma membrane has to be seriously considered as a major molecular mechanism controlling ER Ca^2+^ content and consequently ER-mediated Ca^2+^ release. Finally, it becomes now evident that ER Ca^2+^ dyshomeostasis is significantly associated with AD development and/or progression. The treatment options for AD remain supportive and symptomatic, without attenuation of the ultimate prognosis; thus efforts have still to be made in defining therapeutic approaches targeting ER Ca^2+^ machinery to cure AD (Figure 2). The described ER Ca^2+^ toolkits are enriched in ER–mitochondria contact sites known as mitochondria-associated membranes (MAMs) [240]. Importantly, besides Ca^2+^ tunneling from ER to mitochondria, MAMs impact various cellular housekeeping functions such as phospholipid, glucose, cholesterol and fatty acid metabolism, which are all altered in AD [240,241]. This may further highlight the potential relevance of targeting ER Ca^2+^ handling proteins as an attempt to alleviate both ER and mitochondria dysfunctions associated with AD.

## Figures and Tables

**Figure 1 cells-09-02577-f001:**
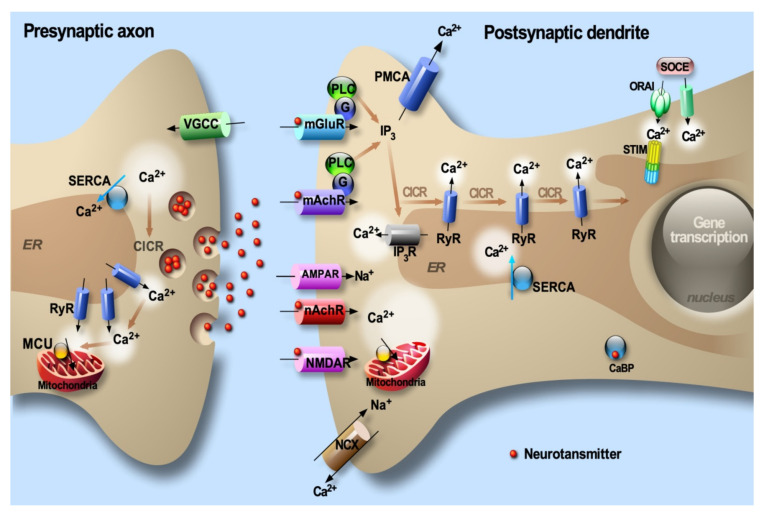
Elevations of intraneuronal [Ca^2+^] are the result of an influx across the plasma membrane and the release from the ER through various channels and receptors. The low intraneuronal Ca^2+^ level is then maintained by the activity of Ca^2+^-binding proteins (CaBP) and involves the sodium-Ca^2+^ exchanger (Na^+^/Ca^2+^) acting in concert with the ATP-dependent Ca^2+^ pumps located at the plasma membrane and the ER. Depletion of ER Ca^2+^ content activates the store-operated Ca^2+^ entry (SOCE) pathway.

**Figure 2 cells-09-02577-f002:**
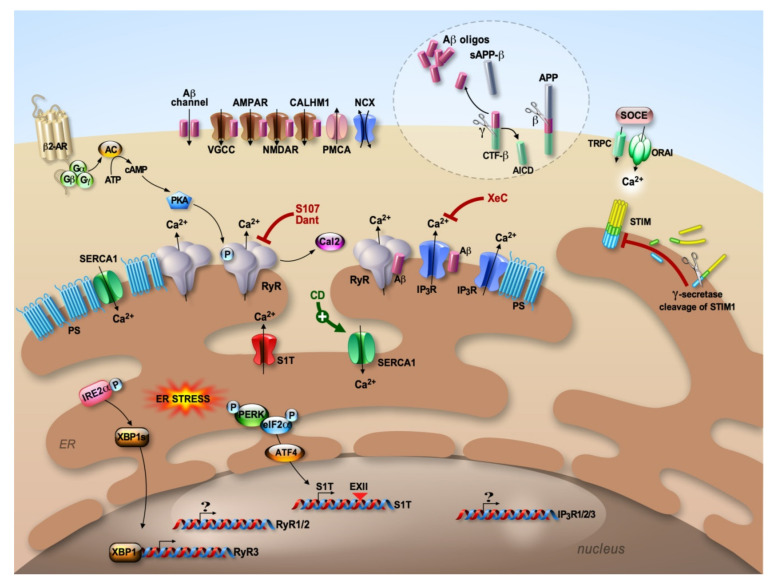
Aβ peptides are derived from the processing of the βAPP (APP) through the amyloidogenic pathway. APP is first cleaved by β–secretase (β), generating APP C-terminal fragment β (CTF-β), which is then cleaved by γ–secretase complex (γ) to produce Aβ and APP intracellular domain (AICD). At the plasma membrane, Aβ form a cation channel and modulate several Ca^2+^ channels (VGCC, AMPAR, NMDAR and CALHM1). ER Ca^2+^ deregulation occurs through presenilin (PS)-associated Ca^2+^ leak and/or enhanced IP_3_R- and RyR-mediated Ca^2+^ release, dysfunctional Sarco-endoplasmic reticulum Ca^2+^ ATPase (SERCA) activity, enhanced expression of S1T driven by ER stress response and enhanced expression and dysfunctional IP_3_Rs and RyRs. SOCE is also deregulated in AD and implicates STIM, ORAI and TRPC. RyR2 macromolecular complex destabilization (PKA phosphorylation and calstabin2 (Cal2) dissociation) is linked to β2-adrenergic receptor activation. Pharmacological stabilization of ER Ca^2+^ content by S107, Dantrolene (Dant) blocking RyRs-mediated Ca^2+^ release/leak, Xestospongin C (XeC) blocking IP_3_R-mediated Ca^2+^ release and CD1163 (CD) activating SERCA provided beneficial effects in reversing several AD-related pathogenic paradigms in vitro and in vivo.

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
