# Peer review of "Alterations of the Endoplasmic Reticulum (ER) Calcium Signaling Molecular Components in Alzheimer’s Disease"

_cells, 2020, doi:10.3390/cells9122577_

Round 1

Reviewer 1 Report

This is a nice review with appropriate figures focussing on important players of calcium homeostasis.

Author Response

We thank the reviewer for her/his positive comments

Reviewer 2 Report

The review by Chamia and Checler provides a very complete and updated view of the present knowledge on the role of ER Ca2+ signaling in the pathogenesis of Alzheimer Disease. The literature is well presented, and the review is useful for a general readership. The topics included have also been object of several other reviews in recent years, but I can say this review is nicely written and well organized and documented, and indubitably, it warrants publication.

Most of the topics related to the connection between ER-Ca2+ signaling and AD are very well described. However, I miss a section devoted to the possible role of alterations in the function of the MAMs in the pathogenesis of AD. In fact, in the last few years, a lot of interest has emerged in the possible role of MAMs in multiple pathological processes, including AD.

As a minor point, sub-indexes and super-indexes should be reviewed throughout, including Ca2+ and some others.

Author Response

We thank the reviewer for her/his positive comments.

The number of studies focusing of MAMs dysfunction in AD is increasingly growing. We prefer to keep the main focus of our current review on ER calcium deregulation. However, we mentioned in the conclusion section lines 465-470, the contribution of MAMs to AD and cite two recent references about this topic.

We also corrected all sub-indexes and super indexes in the manuscript.

Reviewer 3 Report

The article „Alterations of the Endoplasmic Reticulum (ER) Calcium Signaling Molecular Components in Alzheimer’s Disease” by Mounia Chami and Frédéric Checler reviews crucial molecular ER Ca2+ components that contribute to calcium deregulation in Alzheimer’s disease (AD). Moreover, the Authors focus on the potential benefit of targeting ER Ca2+ homeostasis as a tool to diminish AD pathogenesis. The review is very good written, all important aspects  of the described subject have been displayed very precisely.  The review presents also up-to-date literature. In summary, the presented scientific problem is important and interesting. The scientific value the review is very high.

Minor comment:

It would be very interesting to focus a little bit more on physiological function of APP in the regulation of ER calcium levels.

Author Response

We thank the reviewer for her/his relevant comment. We amendthe review with this information in the introduction section lines128-132 and provid two relevant references.